# Deep SE-BiLSTM with IFPOA Fine-Tuning for Human Activity Recognition Using Mobile and Wearable Sensors

**DOI:** 10.3390/s23094319

**Published:** 2023-04-27

**Authors:** Shaik Jameer, Hussain Syed

**Affiliations:** School of Computer Science and Engineering, VIT AP University, Amaravati 522237, India; jameer.sk4u@gmail.com

**Keywords:** data pre-processing, feature extraction, hyperparameters, time series data, wearable sensors

## Abstract

Pervasive computing, human–computer interaction, human behavior analysis, and human activity recognition (HAR) fields have grown significantly. Deep learning (DL)-based techniques have recently been effectively used to predict various human actions using time series data from wearable sensors and mobile devices. The management of time series data remains difficult for DL-based techniques, despite their excellent performance in activity detection. Time series data still has several problems, such as difficulties in heavily biased data and feature extraction. For HAR, an ensemble of Deep SqueezeNet (SE) and bidirectional long short-term memory (BiLSTM) with improved flower pollination optimization algorithm (IFPOA) is designed to construct a reliable classification model utilizing wearable sensor data in this research. The significant features are extracted automatically from the raw sensor data by multi-branch SE-BiLSTM. The model can learn both short-term dependencies and long-term features in sequential data due to SqueezeNet and BiLSTM. The different temporal local dependencies are captured effectively by the proposed model, enhancing the feature extraction process. The hyperparameters of the BiLSTM network are optimized by the IFPOA. The model performance is analyzed using three benchmark datasets: MHEALTH, KU-HAR, and PAMPA2. The proposed model has achieved 99.98%, 99.76%, and 99.54% accuracies on MHEALTH, KU-HAR, and PAMPA2 datasets, respectively. The proposed model performs better than other approaches from the obtained experimental results. The suggested model delivers competitive results compared to state-of-the-art techniques, according to experimental results on four publicly accessible datasets.

## 1. Introduction

Based on device interactions for predicting human activities, HAR is a challenging task. It helps people in various ways in their daily lives [1,2]. Two techniques are used to recognize human behavior: video image recognition and wearable sensors [3,4]. To identify human activities, the camera is utilized by video systems [5]. This approach demands the deployment of expensive cameras and networks, and causes problems due to the scale circumstance, lighting, and background that make detection challenging [6,7]. The next method relies on wearable sensors to identify human movement, specifically gyroscopes, magnetometers, accelerometers, linear acceleration, and other devices that translate human actions into identifiable signals [8,9]. With fewer environmental restrictions, a new dimension is added to moving by the sensor-based method compared to a video-based method, and also gives the user privacy [10,11]. 

Many medical applications use wearable activity detectors for daily fitness tracking [12,13]. Despite this, there is ongoing research to improve the role of inertial sensors in HAR, and the effectiveness of those technologies is still in debate [14]. Data from an inertial sensor time series training for action detection currently needs to utilize deep learning standards, despite the recent focus on deep learning approaches for machine learning in the scientific community [15,16]. Over the last decade, there have been various recommendations for HAR utilizing accelerometers in the literature of conventional machine learning and extensively trained approaches [17]. However, in real-world applications, it is possible that HAR systems cannot change to accommodate new users or situations, which results in low computational efficiency [18]. 

Numerous factors can influence the performance of activity-recognition systems, such as sample frequency and sensitivity range variations across different sensor brands and device positions (for example, pocket, bag, or hand) [19]. Few studies have examined the effects of device and position attributes, despite significant research on the influence of human traits on recognition accuracy [20].

For HAR, we propose a multi-branch SE-BiLSTM with an IFPOA model with different kernel sizes in this research using wearable sensor data. By utilizing both SqueezeNet and BiLSTM advantages, the proposed model improves the performance of HAR. The SqueezeNet model retrieves useful local features, whereas long-term dependencies are managed through the BiLSTMs. Compared to baseline LSTMs and BiLSTMs, BiLSTMs use both past and future information, whereas the baseline LSTMs use only past information once all-time series data have been collected. As a result of the increased context offered, the network can produce more accurate predictions. The different size convolutional kernels are used in this model. In sequential data, several temporal local dependencies are captured by this model. IFPOA is used to optimize the hyperparameters of the BiLSTM network. In multiple human activities’ detection, datasets, including MHEALTH, KU-HAR, and PAMPA2, are used to assess the performance of the proposed model.

The main contributions of the research are:We proposed the Deep SE-BiLSTM-IFPOA for activity classification problems using three public datasets; MHEALTH, KU-HAR, and PAMPA2.First, the input signals are pre-processed for better recognition results. Linear interpolation, segmentation, class relabeling, and one-hot encoding are performed in the pre-processing stage.Then, pre-processed input data are fed to the SqueezeNet model to capture local features. Following, the long-term dependencies among actions are captured by the Bi-LTSM for further improving the HAR identification rate.In the HAR classification process, the IFPOA is used for fine-tuning the hyperparameters of the BiLSTM model for HAR performance improvement.The experimental results are then assessed and contrasted with those of other models using the same dataset. The results demonstrate that the proposed strategy outperforms other cutting-edge techniques.

Sections for the remaining information in this work are as follows: The literature on deep learning-based human activity detection and existing issues are examined in Section 2. Section 3 effectively explains the details of the proposed methodology. Section 4 provides the experimental results and datasets used for the proposed method. Section 5 provides the conclusion of the proposed research.

## 2. Related Works

Table 1 shows the literature on deep learning-based human activity detection and existing issues. Artificial intelligence and deep learning are primarily used in signal-processing techniques for human activity recognition. Several works have already been conducted on this research topic. We reviewed some of the international journals on HAR using deep learning.

Han et al. [21] concentrated on enhancing vanilla convolution in the HAR environment without changing the model architectures. For the activity recognition challenge, a novel heterogeneous convolution is provided. The concept of grouped convolution inspires that. In particular, a low-dimensional embedding is created from down sampled sensor input. The ability of the two filter groups to complement one another is beneficial for expanding the sensor signals’ receptive field for the HAR task. 

Luptáková et al. [22] proposed a transformer model. In the transformer, the self-attention mechanism is used. Within a time series, the various dependencies between signal values are expressed by this integrated model. Correlations between features in time steps are correctly categorized by the transformer model.

In an indoor environment, Khan et al. [23] developed a novel hybrid model using deep learning for monitoring and recognizing human activities. A hybrid model for activity recognition that combines CNN and LSTM is proposed in this research. In this hybrid model, the activities’ features are extracted using CNN, and temporal information is learned using the LSTM network model.

For real-time HAR, the computation-efficient CNN is proposed by Cheng et al. [24] on mobile and wearable devices utilizing conditionally parameterized convolution. The initial challenge for sensor-based HAR is dealing with several channels of time series sensor signals.

A deep neural network-based model is proposed by Dua et al. [25]. The feature extraction is conducted using CNN, and the activities’ classification is performed using Gated Recurrent Unit (GRU). CNN is used for extracting significant action features. Then, GRU layers are used for effectively learning long-term dependencies of data in this research. Hence, the model can capture the diversity of data. The model uses three distinct convolutional filter sizes and has three-head architecture. At different lengths, the local correlations are captured by this model successfully by using several filter sizes.

An attention-based multi-head model is proposed by Khan et al. [26] for HAR. Three lightweight convolutional heads are included in this architecture. Using each head, the features from sensory data are extracted through one-dimensional CNN. The representation ability of CNN is strengthened by inducing the lightweight multi-head model with attention. Each head of the proposed multi-head model is constructed using two convolutional blocks. The salient features are automatically selected by this model and also, it ignores the unimportant features.

To solve the difficulties obtained from the existing HAR models, such as. (i)The generalization capability of the network is increased, (ii) the proposed network model performs effectively for highly complex activities, (iii) the computational cost of the proposed model is decreased, (iv) the more reliable features of the activities are extracted by the proposed network model, (v) it performs very well on long-term HAR in the real-world applications, (vi) the proposed network model does not require additional time for training on enlarged data, (vii) the proposed network model increases the recognition rate of the activities. Therefore we propose a reliable and effective hybrid deep learning-based network model named Deep SE-BiLSTM-IFPOA model. Thus, the methodology and work presented in this research will result in reliable outcomes.

## 3. Proposed Methodology

To recognize human activity, a hybrid Deep SE-BiLSTM model is presented. The signal pre-processing is conducted using a linear interpolation algorithm, sliding window segmentation, and class resampling. The high-level discriminative features that describe an activity’s characteristics are found using SqueezeNet, reducing the effects of outliers and irrelevant sensor data. The BiLSTM uses the features’ gating mechanisms to encode the bidirectional long-range dependencies. The BiLSTM effectively resolves the vanishing gradient and information loss issues. The optimal hyperparameter values are empirically determined by conducting the batch size, BiLSTM merge mode, and hyperparameter tuning on an optimizer. The loss function is reduced by updating the parameters of the network in the training phase. The network’s hyperparameters are updated using an IFPOA optimization algorithm. Three HAR datasets, including an MHEALTH dataset, the KU-HAR dataset, and the PAMPA2 dataset, are tested to compare the performance of the proposed SE-BiLSTM with the existing approaches. The architecture of the proposed methodology is given in Figure 1.

### 3.1. Dataset Description

The activities of humans can be detected using a variety of wearable sensor-based datasets. However, sensor placements, a wide range of sampling rates, recorded activity counts, and sensor counts are used to collect most data. MHEALTH, KU-HAR, and PAMPA2 datasets are used to experiment and verify the proposed approach in this research. Some of these datasets are quite balanced, whereas others are noticeably imbalanced. The datasets utilized in the proposed research are detailed below in brief. 

#### 3.1.1. MHEALTH

Ten volunteers were fitted with sensors at their right, left, and chest handles to monitor their physiological signals and emerging signs (magnet field direction, acceleration, swing rate) while performing 12 physical activities, such as Running (Run), Jumping front and back (JFB), Lying down (Lay), Cycling (Cycl), Knees bending (crouching) (KB), Jogging (Jog), Climbing stairs (CS), Waist bends forward (WBF), Frontal elevation of arms (FEA), Standing still (Std), Walking (Walk), Climbing stairs (CS), and Sitting and relaxing (Sit). All actions are captured at a sample rate of 50 Hz.

#### 3.1.2. KU-HAR Dataset

This collection has many samples, which are classified into 18 classes (activities). Ninety individuals, ages 18 to 34, provided the HAR data. Each participant wore a waist bag with gyroscope and accelerometer sensors from a smartphone for the measurements. The smartphone’s screen faced the participant towards the bag’s left side; 100 Hz is used as the sampling rate.

#### 3.1.3. PAMAP2 Dataset

Researchers from the German Research Center for Artificial Intelligence’s Department of Augmented Vision created the PAMAP dataset; participants aged 27 to 30 years-old are in the data collection. The participants are one female and eight males. The dominant’s ankle, chest, and arm had three Inertial Measurement Units (IMUs) attached and a heart-rate monitor. There are 18 different types of activities in this dataset, including “rope jumping”, “cycling”, “walking”, and more; 100 Hz is used as the sampling rate.

### 3.2. Signal Pre-Processing

In the pre-processing stage, we perform the dataset’s under-sampling and over-sampling process to solve the class imbalance issue. We are then applying linear interpolation operation, segmentation, and class resampling to improve the activity recognition performance. A brief description of the signal pre-processing processes is given below.

#### 3.2.1. Under-Sampling and Over-Sampling

Class mismatches in data categories are typical while gathering information on human activity in the environment. Despite having the same time stamp but different amounts of data, the three sensor datasets correspond to the same time. The quantity of data points varies for the gyroscope, linear acceleration, and accelerometer because each sensor’s sampling rate differs. Resampling the training dataset randomly is one way to solve the class imbalance problem. The imbalanced dataset is randomly resampled by two primary processes: over-sampling (replicating examples of minority class) and under-sampling (excluding samples of majority class). Under-sampling often has an advantage over random oversampling. Random oversampling is enhanced by creating a popular oversampling method called SMOTE (Synthetic Minority Oversampling Technique). The data imbalance is solved by over-sampling with maximum and minimum values. 

#### 3.2.2. Linear Interpolation

We practically used the smartphone’s sensors to gather the KU-HAR datasets. In the right pockets of subjects, the smartphone is placed. During the compilation process, specific utilized data are missed, typically described by NaN or 0. The linear interpolation algorithm is used to solve this problem and this algorithm fills in the missing value.

#### 3.2.3. Segmentation

The sensor data are primarily divided into fixed-size windows using the sliding window approach. For each selected dataset, the proposed model uses a sliding window of size 128 and 50% overlap. The data size for each window of readings is 6 MB, containing nine feature data points. Frames of size (128, N) are used to segment the data, where the numbers of features or frames are represented by N. Figure 2 illustrates the process of data pre-processing.

#### 3.2.4. Class Relabeling and One-Hot Encoding

Additionally, one-hot encoded labels are created from the output labels. Four unique labels are used to code the activity windows of the total dataset. 

### 3.3. Deep SE-BiLSTM-IFPOA Architecture

Combining two deep learning architectures for HAR, a Deep SE-BiLSTM-IFPOA is first proposed using sensor data. In this research, we offer an end-to-end structure with a feed-forward neural network-based SqueezeNet model for extracting the spatial features automatically related to human appearance. From the sensor data, the long-term temporal actions are captured by using the BiLSTM network model. Effective new encoded feature representations are generated by combining the Deep SqueezeNet and BiLSTM networks for accurate HAR. The best weight initialization is ensured by reducing the required training time and avoiding the overfitting problem of the network. We used IFPOA to fine-tune the parameters of the proposed network model. Finally, to improve the model’s ability to distinguish between relevant characteristics and avoid the model overfitting, we artificially enlarged our datasets by applying some pre-processing and segmentation techniques. We have thoroughly tested our system by validating it on four publicly accessible datasets, namely MHEALTH, KU-HAR, and PAMPA2. This has allowed us to highlight the advantages of our approach. The results (99.98% on MHEALTH, 99.76% on KU-HAR, and 99.54% on PAMPA2) are competitive with state-of-the-art performance. This confirms the effectiveness and efficacy of the proposed methodology. The structure of the proposed Deep SE-BiLSTM model is displayed in Figure 3. Table 2 reveals the details of the proposed Deep SE-BiLSTM model. The layer names and each layer’s hyperparameter settings are listed in this table. The proposed Deep SE-BiLSTM model has a total of 12 layers.

Several hidden layers are used to construct the proposed Deep SE-BiLSTM model, and the number of hyperparameter settings is more customized in this model than in the CNN-LSTM model. The proposed network model is a better solution that should be used to manage large time series sequences and prevent information loss compared to the unidirectional LSTM model. RCNN-UNet architecture has several layers and a complicated structure due to this, and it requires more training time than the proposed Deep SE-BiLSTM model. The network scratch occurs in the hybrid CNN-GRU model for the task of activity recognition when compared to our proposed Deep SE-BiLSTM because hidden layers are reduced, and regularization parameters help us to overcome this scratch problem. 

#### 3.3.1. Feature Extraction Using SqueezeNet

The significant features are extracted from the input HAR signals using the deep learning-based SqueezeNet model [27]. For a wide range of computer vision applications, the SqueezeNet model is the most effective preferable model. This is because several hidden layers are used to construct the SqueezeNet model, and the number of hyperparameter settings is customized in the SqueezeNet model. Due to these characteristics, the feature extraction network learns the internal representation of multiple signal dimensions for the feature learning process and uses the uniform process on 1D signal data in situations like time series. 

In the feature extraction network model, three pooling layers and three convolution layers (one pooling layer follows each convolution layer) are present and eventually generate several feature maps with action features. The filter’s receptive field size is represented by K×K notation, s denotes the stride size, and the feature map length is defined by l, respectively. Figure 4 shows the structure of the SqueezeNet model.

SqueezeNet, consisting of a convolution layer and a pooling layer, receives HAR signals as input. Using the convolution kernel, the feature mapping of the signal is obtained by applying the convolution operation to the input signal in the convolutional layer. From the convolution layer’s feature map, the local features are extracted by the pooling layer using a sampling operation for reducing the number of parameters and size of neurons. A deep structure is created by combining the convolution layer with the pooling layer, and this automatically extracts the action feature information from the original action data. It provides several feature maps with action features as its final result. Convolution combines signals from multiple adjacent frames with a two-dimensional convolution kernel. The three convolution layers’ respective convolution kernel numbers are 18, 36, and 72.
(1)$$f(x)=\max\;(0,x)=\left\{\begin{array}{c} 0,\;\;\;\;\;\;x<0,\\ x,\;\;\;\;\;\;x\geq 0. \end{array}\right.$$

Further, we have:(2)$$f^{\prime }(x)=\left\{\begin{array}{c} 0,\;\;\;\;\;\;\;\;\;x<0,\\ 1,\;\;\;\;\;\;\;\;\;\;x\geq 0. \end{array}\right.$$

In our experiment, 2 × 8, 2 ×18, and 2 × 36 convolution kernels are used, with a step size of one. Convolutional processing may not be possible if the filter cannot handle the information in a specific direction. The reducing signal data are avoided by introducing the padding parameter, setting it to “SAME,” and adding 0 to the edge of the input signal. A non-linear activation function will be applied topically to the output in the convolution layer after the convolution operation and then creates the convolution layer’s output. The ReLU function and sigmoid function are some of the well-known activation functions. In this feature extraction network, the negative data value is changed to 0, and the positive data value is unchanged using the ReLU function. The retrieved features can more clearly describe the positive value greater than 0 after a non-linear processing operation. As a result, CNN’s convolution layer uses the ReLU activation function.

The pooling layer reduces the number of parameters and feature mappings. Average pooling and maximum pooling are popular pooling techniques. According to several studies, the maximum pooling approach is also well-suited for sensor-based human behavior recognition. As a result, all SqueezeNet pooling layers in this research use the maximum pooling strategy.

The squeezing and extension processes both utilize the ReLU units. While maintaining the exact size of the features, the squeeze operation compresses the depth, and the expansion increases the depth. In the input tensor’s depth dimension, the expansion outputs are layered using the concatenate action. Assume the channels and feature maps are defined by *C* and *FM*. The squeeze operation’s output layer *f{y}*, *w* can be represented as with the kernel, i.e.: (3)$$f\{y\}=\overset{FM}{\underset{fm1=1}{\sum }}\overset{c}{\underset{c=1}{\sum }}w_{c}^{f}x_{c}^{fm1}$$

Here, $f\{y\}\in R^{N}$ and $w\in R^{c\times 1\times FM2}$. The squeeze outputs define a weighted combination of the different tensors’ feature maps. A down-sampling operation is executed by the max-pool layers in the network along the spatial dimensions. The class maps with features are converted into one value by the average global pool. The softmax activation function toward the network’s end generates the multiclass probability distributions.

#### 3.3.2. Dropout Layer

The overfitting problem is avoided among the convolutional layers by inserting three dropout layers with a 0.2 dropout rate (r). 

#### 3.3.3. Max-Pooling Layer

The max-pooling layer must compute each time series vector’s maximum value. The stride size moves the spatial window. The stride size is 1, and the pool size is 8 in the proposed model. Given that the padding is the same across the system, the formula below is used to determine the output shape:(4)$$outputshape=\frac{input\;shape\;}{stride}$$

#### 3.3.4. HAR Using BiLSTM

The feature vectors are sent to the Bi-LSTM network from the convolutional layer. By analyzing long-term patterns in human activity sequences and the effects of initial dependencies, LSTM solves the prevalent vanishing gradient problems in Recurrent Neural Networks [27]. The proposed BiLSTM model uses the hidden state to capture the information it previously perceived from the time series. 

Unidirectional LSTM only works with time series in the single path on conserved data, either past or future. Real-world human activity is continuous, and time series are produced continuously. BiLSTM is a better solution that should be used to manage large time series sequences and prevent information loss. Compared to the unidirectional LSTM model, the proposed BiLSTM is different. Instead of one LSTM layer, two LSTM layers are used by BiLSTM for training—the time series used by BiLSTM flow in both directions, including backward and forwards. As a result, information from the past and the future might be retained, providing the proposed network with more contexts. Consequently, the HAR task might be learned more quickly and thoroughly. 

Several gates are included in the LSTM internal architecture, including an output gate, forget gate, and input gate. Each gate controls the flow of information toward the final output by processing the previous gate inputs and forwarding them to the next gate. A sigmoid activation function often contains all gates; for example, information is updated by the responsibility of the input gate. Previous cell state *CS_t_*_−1_ and input data from the input gate *IN_t_* are processed by the forget gate; when necessary, it also deletes data from the current state *CS_t_*. whereas for the next LSTM unit, the output gate *OP_t_* forwards the final output. The output gate also holds the output value for the next sequence prediction. Using the softmax activation function, the current input value *x_t_* and previous cell state *CS_t_*_−1_ are estimated by recurrent unit *CS_t_*. While the softmax of *CS_t_* and the scalar product of *OP_t_* can be used to get the *h_t_* value. Finally, sending *h_t_* to the softmax classifier will yield the final result. The gates’ operations mentioned above can be stated mathematically as follows:(5)$$f_{t}=\Phi \;({\hat{W}}_{f}.\;[h_{t-1},\;\;x_{t}]+B_{f})$$
(6)$$IN_{t}=\Phi \;({\hat{W}}_{i}.\;[h_{t-1},\;\;x_{t}]+B_{i})$$
(7)$$CS_{t}=\;Soft\max\;\left({\hat{W}}_{CS}.\;[h_{t-1},\;\;x_{t}]+B_{CS}\right)$$
(8)$$CS_{t}=f_{t}\;x\;CS_{t-1}+\;\;IN_{t}\;x\;\;CS_{t}$$
(9)$$OP_{t}=\Phi \;({\hat{W}}_{o}.\;[h_{t-1},\;\;x_{t}]+B_{o}$$
(10)$$h_{t}=OP_{t}x\;\;softmax\;\left(\phi \left(CS_{t}\right)\right)$$
(11)$$Output=softmax\;\left(h_{t}\right)$$

The states of the BiLSTM are represented by *h_(t)_* and *g_(t)_* in the classification model. *h_(t)_* moves forward, and *g_(t)_* moves backward through the ordered sequence, where t=1 ,2 ,3 , . . . ,m.
*h_(t)_* and *g_(t)_* are concatenated by obtaining the output unit *V_(t)_*. The past and the future of the sequence are represented by *V_(t)_* but are most sensitive to the inputs of the current. The calculation of an output vector *V_(t)_* is:(12)$$V_{\left(t\right)}=f(h_{\left(t\right)},\;g_{\left(t\right)})$$
where the two output sequences are combined using the function *f*, it can be a multiplication function, an average function, a summation function, or a concatenating function. The following vector can represent the final result of a BiLSTM layer:(13)$$V_{m}=f(h_{m},\;g_{m})$$
where the predicted sequence is represented by *V_m_*; such a network helps determine the label of the patch sequence because it can summarize a sequence using only the final output vector. The cross-entropy loss function *L* is utilized at the end to back-propagate information through forward *h* states and backward *g* states to train this network. The weight updating is conducted after forward and backward passes. Each set of sequences is sent to the BiLSTM system once, and the sequence class is indicated by the expected output. The prediction is obtained using a softmax classifier. 

#### 3.3.5. Classification Layer

From the Bi-LSTM layers, the retrieved feature vector is then compressed and fed into the classification layer. The fully connected layer is used for activity classification. The classification layer returns all activity classes’ predictions. The activity classification’s probability distribution is measured by applying the softmax activation function (σ): (14)$$\sigma {\left(\vec{Z}\right)}_{i}=\frac{e^{z_{i}}}{\sum_{j=1}^{K}e^{z_{j}}}$$
where the input vector is represented by $\vec{Z}$, the input vector components are described by *Z_i_*. Normalization expression is represented by $\sum_{j=1}^{K}e^{Z_{j}}$ with all outputs being scaled to a range of 0 to 1. Consequently, an effective probability distribution is established.

#### 3.3.6. Hyperparameter Optimization Using IFPOA

The network hyperparameters are tuned to determine the optimal values in model settings. Network settings of the optimizer are adjusted and analyzed, affecting model performance and training for reducing the loss function while attempting to approximate the best results. In network training, network parameters are updated to minimize the loss of function. However, the optimizer used significantly impacts the size and direction of the parameter adjustment. Learning Rate and Regularization are mainly used to evaluate the optimizer’s effectiveness. A loss function with a learning rate that is either too high or too low either fails to converge or falls within the range of the local minimum rather than the absolute minimum. Regularization keeps the model from overfitting without preventing the significant weighting components from making correct predictions. Therefore, the classifier’s generalization ability improves when exposed to new data. The experiments make use of the IFPOA optimization method.

Using IFPOA optimization techniques, the proposed network model’s weight variation is optimized. Based on the natural pollination of flowers, the IFPOA was developed. When pollinating flowers, pollen is frequently moved from a plant’s male to female components. Pollination can occur in numerous ways, including cross-pollination and self-pollination. For pollen transferring, the biological process of cross-pollination needs the assistance of the wind, birds, insects, and water. The objects of global search are performed by using biotic pollinators with the help of levy flight, and objects of local search are achieved by using abiotic pollinators. This represents the switching probability and determines both biotic and abiotic pollination. The following equation represents the flower consistency: (15)$$e_{i}(t+1)=e_{i}(t)+L(e_{i}(t)-q^{*})$$
where the present best solution is represented by *q**, pollen, i along vector *e_i_* at iteration *t* is indicated by the *e_i_(t)*, and levy flight is denoted by *L*. 

Network convergence and accuracy are heavily reliant on the hyperparameter. Based on CNN, network hyperparameters were selected. The typical CNN training hyperparameters are the number of epochs, learning rate, regularization coefficient, and momentum. Based on the previous weight updates, the based on the previous weight updates and the learning rate modifies the network speed. The number of epochs is represented by the learning algorithm, which updates the network several times with the entire dataset. The over-fitting is avoided by regularization. Consequently, the hyperparameters are optimized to create these configurations. Due to this, the network is better able to obtain accurate results.

## 4. Results and Discussion

The training processes’ experimental settings and results of the experiments using the proposed approach are described in this section. The proposed model performance is compared by conducting several experiments. For activity recognition, three cutting-edge datasets are used for assessing the developed hybrid deep neural network-based model (Deep SE-BiLSTM-IFPOA). The proposed model’s performance is validated and tested using MHEALTH, KU-HAR, and PAMPA2 datasets.

A system with 8 GB of RAM and an Intel Core i5 processor is used in this research to experiment. The experimental analyses are compiled using Google COLAB server software. Automatic differentiation for parameter inference and network training is conducted by using Tensorflow as a Python module. 

### 4.1. Evaluation Measures

Class mismatches in data categories are prevalent when gathering information on human behavior in the environment, and this research’s datasets are no exception. Using the overall classification accuracy, the results will be highly accurate if each instance is classified by the modal into a majority class and evaluates the model output. The false negatives and false positives are considered by the F1-score, which also includes two metrics called “recall” and “precision,” which are determined by the overall number of correctly identified samples. The evaluation criteria used in this research are then briefly described:(16)$$Accuracy=\;\frac{TP+TN}{TP+TN+FP+FN}$$
(17)$$\mathrm{Pr}ecision=\;\frac{TP}{TP+FP}$$
(18)Recall=TPTP+FN
(19)F1−Score=2  ×Precision×RecallPrecision+Recall

Positive generally implies recognized, whereas negative means typically disregarded. Accordingly, in the equations mentioned above, *FN* (false negative) denotes instances that were incorrectly rejected, *TN* (true negative) denotes samples that were correctly rejected, *FP* (false positive) denotes instances that were wrongly identified, and *TP* (true positive) denotes instances that were successfully identified.

The hyperparameters are optimized by using the IFPOA optimizer. Over data validation, the proposed model attains higher validation accuracy. The hyperparameter settings are selected at random from the given ranges by the search method. The model’s accuracy is gradually increased by manually adjusting these intervals; ultimately, the best possible combination was selected (see Table 3).

### 4.2. Performance Results on the MHEALTH Dataset

On each dataset, 80% of the data are used for training, and 20% is used for testing for experiment analysis of the proposed model. Totally, 12 activities are present in the MHEALTH dataset: Standing still, Sitting and relaxing, Lying down, Walking, Climbing stairs, Waist bends forward, Frontal elevation of arms, Knees bending (crouching), Cycling, Jogging, Running, Jumping front and back. Table 4 shows the performance results of the proposed model on the MHEALTH dataset. It offers the proposed model’s class-wise precision, recall, and F1 score. These results demonstrated that our proposed Deep SE-BiLSTM-IFPOA was adequately trained and is capable of accurately recognizing human activities. In the MHEALTH dataset, a 99.98% recognition rate is achieved by our proposed model, and this dataset includes 12 activities. We can observe a significant improvement in the performance classification results. To reach the final iteration, 0.9063 s timing is used in the entire training process. The input signals are fed into the 224 × 224 × 3 input layer during the training phase, followed by the two convolution layers for feature extraction. During the training phase, the validation accuracy is improved by selecting hyperparameters. The small sample of the validation set’s signals is employed to determine the class-wise performance. The proposed deep learning model effectively performs HAR classification. 

The confusion matrix values are used for assessing the proposed work’s performance for each class. These values are evaluated using the following metrics: accuracy, sensitivity, and specificity. Figure 5 demonstrates the confusion matrices of HAR classification using MHEALTH. 

Training and validation accuracy and training and validation loss of the proposed model on the MHEALTH dataset are given in Figure 6a,b. The X-axis represents the predicted values (system output), while the Y-axis represents the actual labels (ground truth). The average precision, recall, and F1-score of the 12 involved activities in MHEALTH are 96.95%, 99.09%, and 99.13%, respectively. The proposed Deep SE-BiLSTM-IFPOA achieves the highest accuracy for HAR classification.

The performance comparison of recent research using the MHEALTH dataset is shown in Table 5. Gholamrezaii et al. [28] used 2D-CNNs for HAR, and 99.88% recognition rate was achieved. Jain et al. [29] presented a CNN-LSTM ensemble model for the lower extremity activities recognition task. A relatively low-performance measure such as precision and recall is obtained in this model due to a class imbalance, but the overall accuracy is increased. Dua et al. [30] present the ICGNet based on DNN for HAR. The deep transfer learning model proposed by Varshney et al. [31] achieved an accuracy of 98.63% which is 1.35% less than that of Deep SE-BiLSTM-IFPOA. This is comparatively high compared to the 235,692 parameters employed in the proposed model (Deep SE-BiLSTM-IFPOA). In comparison to the CNN-GRU model of Nafea et al. [32], 0.6% is improved by the proposed model. The proposed Deep SE-BiLSTM-IFPOA model architecture achieves better performance on the MHEALTH dataset in accuracy compared to the predecessor architecture of deep learning models. The performance comparison graph on the MHEALTH dataset is given in Figure 7.

### 4.3. Performance Results on the KU-HAR Dataset

Table 6 shows the statistical evaluation of the results of the proposed model on the KU-HAR dataset. These results demonstrated that the proposed Deep SE-BiLSTM-IFPOA is appropriately trained and is capable of accurately recognizing human activities. The overall accuracy of the proposed model using the KU-HAR dataset is 99.78%. Values of precision, recall, and F1-score are presented for each activity. The precision varies between 0.945 and 1.0 for talk-sit activities and table tennis. The confusion matrix shows that, despite the appearance that table tennis activities cannot be confused, the site activity is usually mistaken for the talk-sit action. Talking has little to no impact on the gyroscope or accelerometer data.

Figure 8 shows the proposed model’s confusion matrix for the KU-HAR dataset. The confusion matrix shows the diagonal’s most significant numbers in dark red. This demonstrates that the actual and predicted labels are generally consistent. The proposed Deep SE-BiLSTM-IFPOA had an average prediction success rate of 99.76%. Both classifying one activity that was present throughout the entire time series and combining two activities into a one-time series are successfully handled by the proposed model. From the time series, removing every second signal measurement had no impact on the predictions’ robustness. The performance of the proposed model predictions is enhanced by developing a signal pre-processing process.

The training phase was conducted with 26 epochs and a 0.01 learning rate for better results. Each epoch used a total of 31 iterations in the experiment analysis. The training and validation accuracy and training and validation loss graph for HAR classification on the KU-HAR dataset are given in Figure 9a,b. For 18 involved activities, the average F1-score, recall, and precision in KU-HAR are 99.18%, 99.00%, and 98.32%, respectively. The proposed Deep SE-BiLSTM-IFPOA achieves the highest accuracy of 99.76% for HAR classification.

The comparison results of the proposed model with existing HAR research on the KU-HAR dataset are given in Table 7. Dahou et al. [33] utilize CNN-BAOA for the HAR process, the CNN model is used for significant feature extraction, and BAOA is used for selecting the most relevant features of activities. It achieves an accuracy of 96.8%. Kumar et al. [34] provide a Recurrent HAR model for HAR, reaching an overall accuracy of 98.98%. This method classifies skin cancer as benign or malignant. Luptáková et al. [22] also achieved an accuracy of 99.2% for HAR classification using the transformer method. While compared to the prior deep learning models, the proposed Deep SE-BiLSTM-IFPOA classification model achieves higher classification accuracy and effectively recognizes human activities. Figure 10 shows the performance comparison on the KU-HAR dataset.

### 4.4. Performance Results on the PAMPA2 Dataset

For experiment analysis, the network training is conducted using 80% of the data, and network testing is performed using 20% of the data. The proposed scheme’s confusion matrix on the PAMAP2 dataset is shown in Figure 11. On the PAMAP2 dataset, we can observe that all models have a more challenging time detecting comparable actions. There is a higher probability of misunderstanding while doing certain activities than others, including ironing and standing, ironing and cycling, and ironing and cleaning. The activity pairs like ironing and cycling, ironing and cleaning, and standing and sitting are effectively recognized by the proposed hybrid network model (SE-BiLSTM-IFPOA). For each of the three pairs of activities, the fewest confusion samples are obtained by the proposed model. There are fewer confusion samples in the proposed network for ironing and standing. The highest recall is obtained for all activities by the proposed network model.

The training and validation accuracy and loss curve of the proposed model on PAMAP2 under 50 epochs are shown in Figure 12a,b. After 20 epochs, each curve’s fluctuation is minimal, and the accuracy rises with the number of epochs. The proposed model reaches an overall accuracy of 99.54%. For 12 involved activities, the average F1-score, recall, and precision in the PAMPA2 dataset are 99.27%, 99.21%, and 99.02%, respectively.

The comparison results of the proposed model with prior deep learning research are given in Table 8. CNN-D was developed by Jin et al. [35], the combination of LSTM, CNN, and MLP, and the developed CNN-D model reveals higher effectiveness for HAR. The proposed model’s accuracy improves by 4.82% when compared to CNN-D. The accuracy rises by 1.92% compared to [24], which used Condconv for the basic convolution layer. Zhou et al. [36] utilize the MLWP model for the HAR process. According to the properties of static activity signals with minor fluctuations in this MLWP process, slope-area (SA) and threshold segmentation (TS) techniques are used, and it achieved an accuracy of 95.93%. M-branch G-CNN is proposed by Li et al. [37], the most important features are selected by introducing a dual attention mechanism that combines spatial attention and channel attention, and it achieves an accuracy of 97.35. Feng et al. [38] proposed a CNN-Bi-LSTM model to detect the real-time building occupancy from publicly available AMI data with 90% occupancy detection accuracy. The proposed network model’s accuracy is also considerably higher than those of existing HAR techniques. The PAMAP2 dataset shows that the proposed model is the most accurate, with a 99.54 accuracy rate from Table 8. Effective classification results are maintained by the proposed Deep SE-BiLSTM-IFPOA model with less computational power in conclusion. The performance comparison graph on the PAMPA2 dataset is given in Figure 13.

### 4.5. Discussion

All action-based human activity problems are addressed by the proposed model in this research such as heterogeneous, interleaved, concurrent, complex, and sequential activities. For HAR, the goal of the research is to develop a general approach using deep learning and assess the effectiveness using a wide range of HAR datasets. The deep learning-based hybrid model named Deep SE-BiLSTM with IFPOA fine-tuning is proposed in this regard. Using three public datasets KU-HAR (heterogeneous activities), PAMAP2 (interleaved, concurrent, and complex activities), and MHEALTH (sequential activities), extensive experiments are performed for evaluating the proposed model.

The proposed hybrid classification network model has the main benefit of preventing overfitting and having no negative effects on network performance due to the classification process. The proposed model successfully extracts characteristics that represent the inter-scale variability of human activities, improving recognition performance and resulting in an improvement in classification performance. For HAR, the proposed method performs more effectively than the recent existing methods on MHEALTH, KU-HAR, and PAMP2 with accuracies of 99.98%, 99.76%, and 99.86%. The human activities are automatically diagnosed and pre-screened by the proposed hybrid model (Deep SE-BiLSTM-IFPOA) due to its encouraging classification accuracy.

While compared to other widely used deep learning models, results indicating the proposed hybrid deep learning network model frequently appears to have better representation capacity in other applications provided evidence in support of this goal. More than one million parameters are effectively scaled by the proposed model. Accuracy and processing time are compromised by the proposed Deep SE-BiLSTM-IFPOA model and performance on HAR is increased by applying the proposed model to several network systems. The proposed model is less complex than other approaches from the experimental results in terms of the overall number of parameters utilized. For real-time embedded implementation, the proposed model is appropriate for the purpose because a lightweight model is preferred. Moreover, the approach uses fewer computational resources and has smaller flops. 

Activity recognition is negatively impacted by activity similarity, and is reduced by the proposed model from the performance of the confusion matrix on MHEALTH, KU-HAR, and PAMP2, and also the proposed model enhances the accuracy of sensor-based HAR. Our method performs better than existing state-of-the-art methods when compared to the results of previous studies. The proposed model achieved 99.98%, 99.76%, and 99.54% of accuracies for the MHEALTH, KU-HAR, and PAMAP2 datasets, respectively. The information on the application of the proposed approach and the potential for further research is provided by the experimental results. 

### 4.6. Ablation Study

Ablation research with two models is examined to thoroughly examine the suggested Deep SE-BiLSTM model, including the CNN-BiLSTM subnet and unidirectional HiHAR. The first method is implemented to test the effectiveness of learning local spatial and temporal variables by eliminating the global BiLSTM from the HiHAR model. The model demonstrates how the adjacent windows affect the current window’s activity prediction by merely using sensor data from a single window to identify the relevant activity. The global BiLSTM layer is swapped out for an LSTM layer to implement the unidirectional HiHAR, which explores how the current classification is impacted by future data. Table 9 presents the findings of the ablation study.

On all datasets, the subnet outperforms the LSTM-CNN model and performs competitively with the proposed Deep SE-BiLSTM model. This finding indicates that the model’s ability to learn data representations is enhanced using 2D convolutional kernels and a bidirectional recurrent network. The average accuracy of the subnet is 2% less accurate than the suggested model due to the model’s view being constrained by just using the current window for activity prediction. An average of 97.09% accuracy is achieved by the unidirectional HiHAR when only the past data are provided, which is 2% less than the suggested HAR model. The latency is cut in half to the window size (1.28s) because the model does not need any future data. As a result, unidirectional HiHAR can be implemented with high performance when compared to other cutting-edge models in the case that a rapid response is required.

There are some errors in the subnet’s standing and sitting categories, despite the fact that it does an excellent work of recognizing and tracking the majority of activities. Among sitting and standing, the angular velocity and acceleration data are occasionally comparable to one another because the phone was mounted at the user’s Waist in the KU-HAR dataset. The suggested Deep SE-BiLSTM model, however, has more confidence in separating the two activities by utilizing past and future information. This result has further supported the association between human activities. The model’s ability to predict activity is now more specific due to the use of context information from adjacent windows.

## 5. Conclusions

A hybrid deep learning model for sensor-based human activity recognition, known as the Deep SE-BiLSTM-IFPOA, is presented in this research. The features of the sensor data are encoded by using SqueezeNet layers in the proposed research. For protecting the class-specific attributes, the long-range dependencies are then encoded in the features by the Deep BiLSTM layer. The IFPOA is used for hyperparameters’ optimization of the network. The MHEALTH dataset, the KU-HAR dataset, and the PAMAP2 dataset are used to compare the performance of the proposed Deep SE-BiLSTM-IFPOA model. The performance of the proposed Deep SE-BiLSTM-IFPOA model is more effective than the previous HAR approaches in terms of 99.98% accuracy on the MHEALTH dataset, 99.76% accuracy on the KU-HAR dataset, and 99.54% accuracy on the PAMAP2 dataset. The HAR is significantly enhanced by the proposed Deep SE-BiLSTM-IFPOA since deep learning models can automatically learn from new time series. Real-time recognition of elderly security systems and health issues will be the focus of our future research.

## Figures and Tables

**Figure 1 sensors-23-04319-f001:**
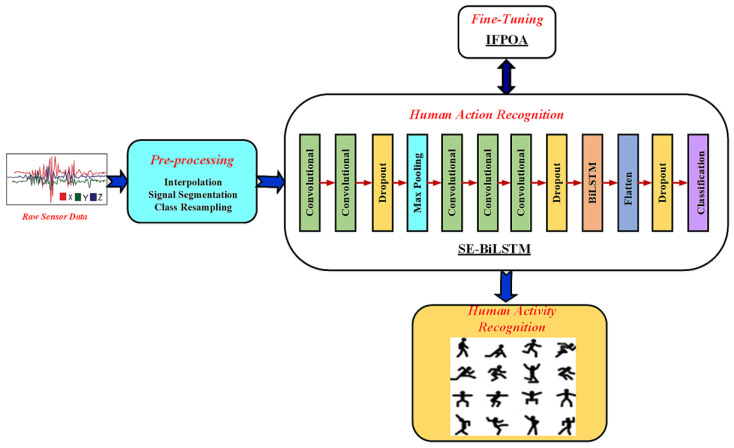
The schematic diagram of the proposed methodology.

**Figure 2 sensors-23-04319-f002:**
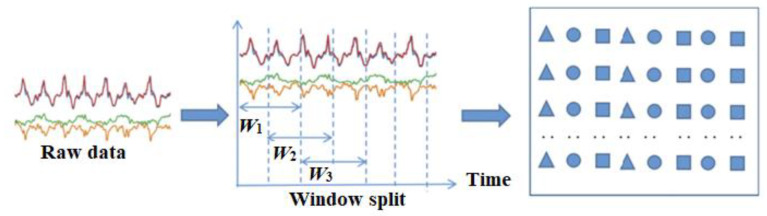
Data pre-processing process structure.

**Figure 3 sensors-23-04319-f003:**
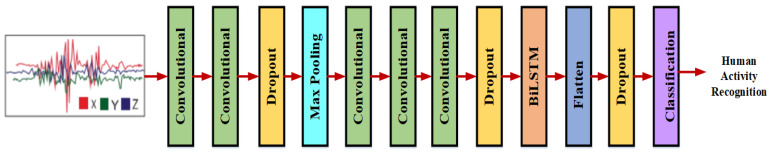
Proposed model architecture.

**Figure 4 sensors-23-04319-f004:**
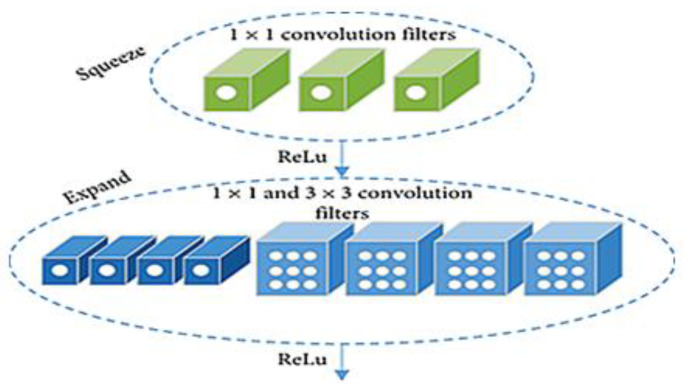
The network architecture of the SqueezeNet feature extraction model.

**Figure 5 sensors-23-04319-f005:**
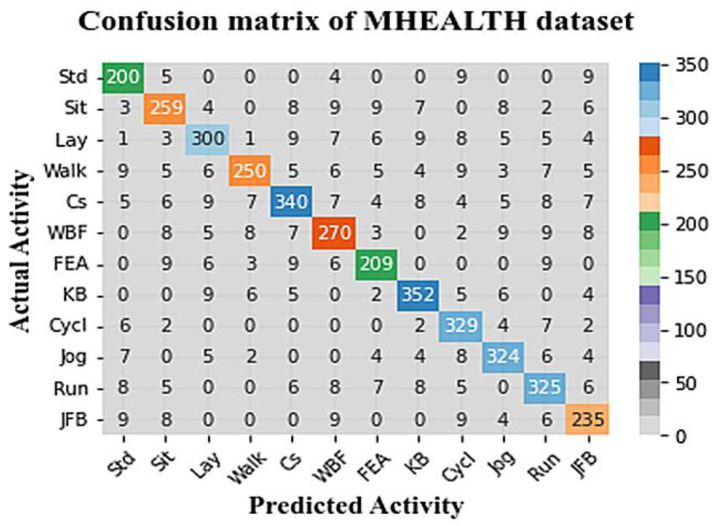
The confusion matrix of the proposed model using MHEALTH.

**Figure 6 sensors-23-04319-f006:**
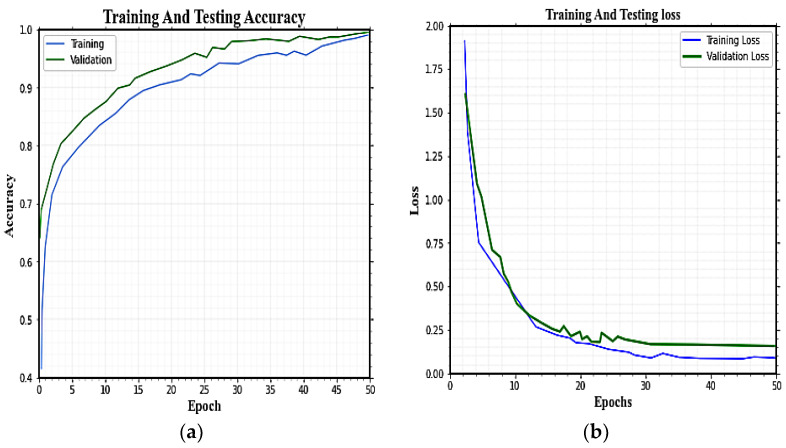
Accuracy and loss curve for both training and validation using MHEALTH.

**Figure 7 sensors-23-04319-f007:**
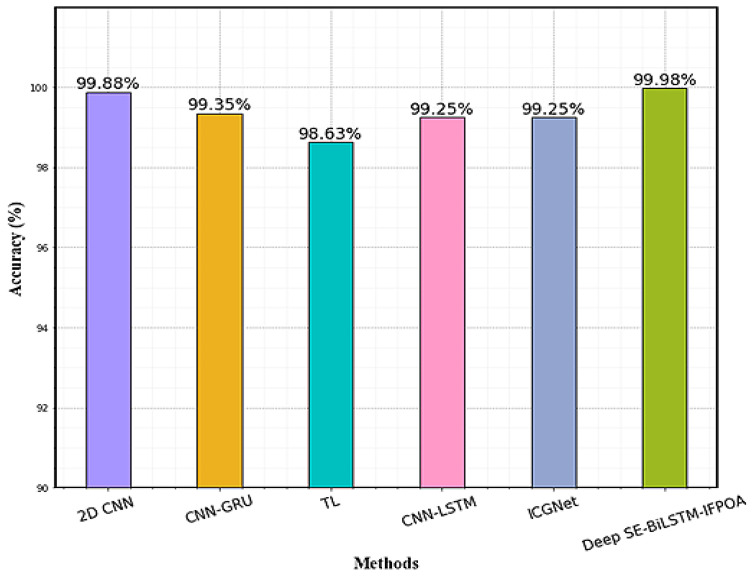
Performance comparison graph on MHEALTH.

**Figure 8 sensors-23-04319-f008:**
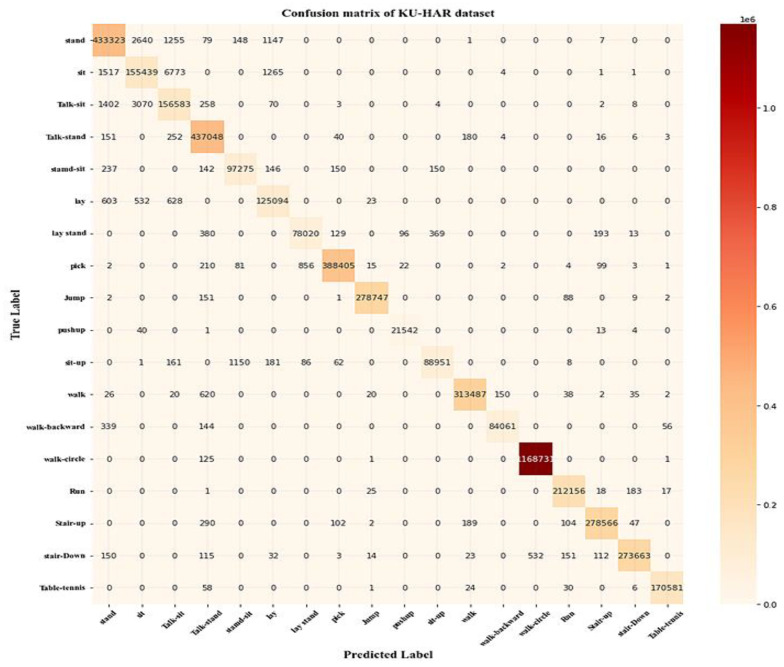
The proposed model’s confusion matrix for the KU-HAR dataset.

**Figure 9 sensors-23-04319-f009:**
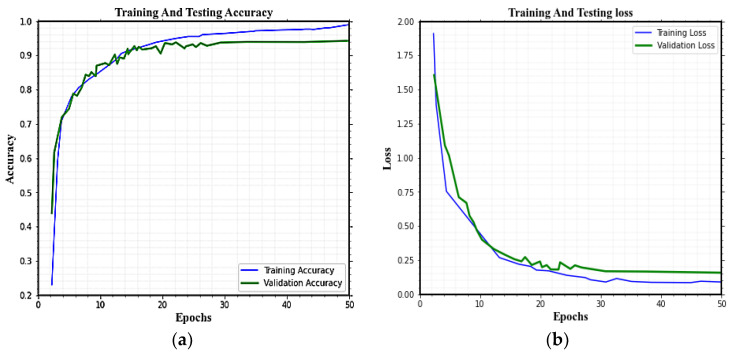
The training and validation performance (accuracy and loss) of HAR classification using the KU-HAR dataset.

**Figure 10 sensors-23-04319-f010:**
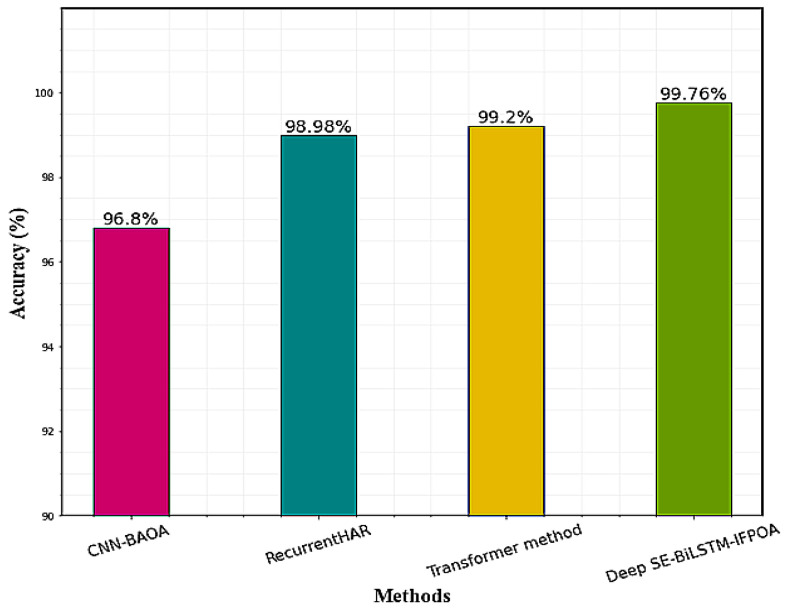
Performance comparison graph on KU-HAR.

**Figure 11 sensors-23-04319-f011:**
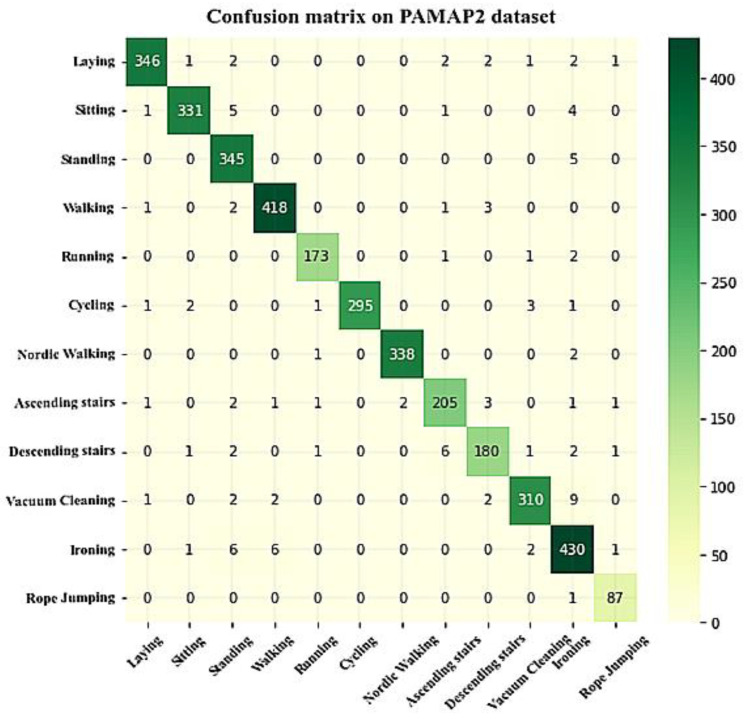
Confusion matrix of the proposed model on the PAMPA2 dataset.

**Figure 12 sensors-23-04319-f012:**
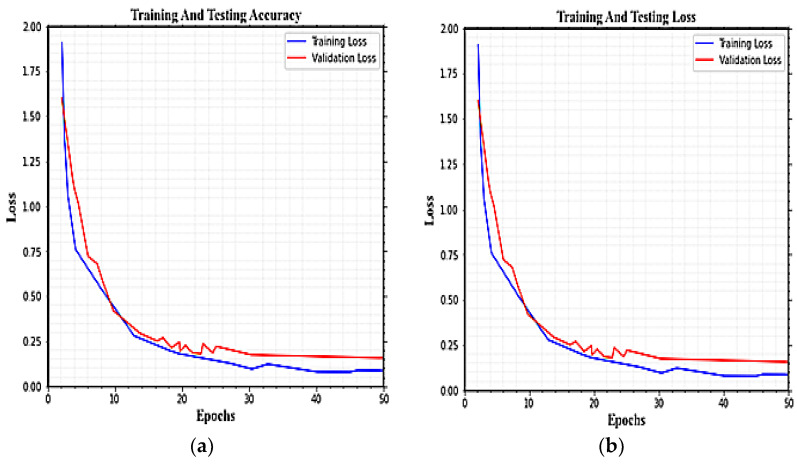
The training and validation performance (accuracy and loss) of HAR classification using the PAMAP2 dataset.

**Figure 13 sensors-23-04319-f013:**
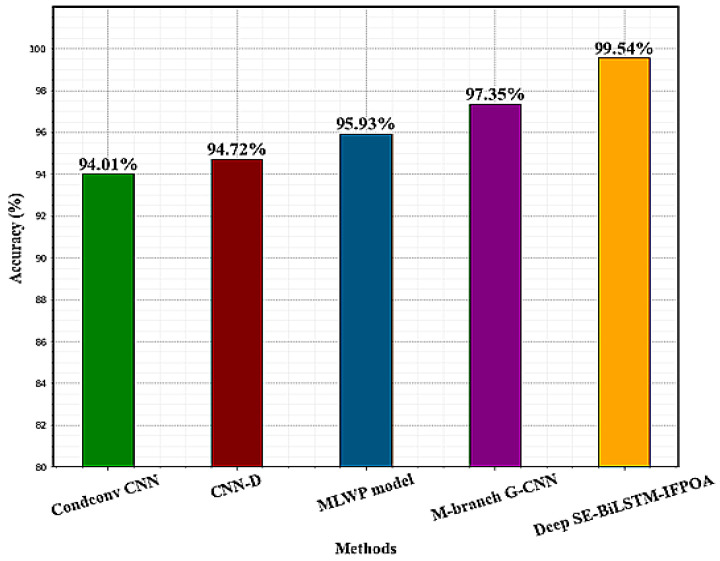
Performance comparison graph on the PAMPA2 dataset.

**Table 1 sensors-23-04319-t001:** Literature Survey.

Reference	Year	Model	Benefits	Difficulties
Han et al. [21]	2022	Heterogeneous two-stream CNN network	(i) At various time scales, this model is more helpful for extracting better activity features. (ii) It improves performance gain significantly without changing the network’s architecture across various HAR application areas.	(i) The minimum tradeoff between privacy protection and recognition performance (ii) The recognition rate is decreased
Luptáková et al. [22]	2022	Transformer model–self-attention mechanism	(i) It offers an outstanding opportunity for stopping privacy leaks. (ii) Correlations between features in time steps and information from time series are correctly categorized.	(i) It does not provide efficient performance for enlarged data (ii) Requires additional time for training for large datasets
Khan et al. [23]	2022	CNN-LSTM	(i) Recognizes the several activity types effectively (ii) It enhances a person’s physical and mental health better	It performs poorly on long-term HAR in the real world.
Cheng et al. [24]	2022	CNN	It provides higher classification accuracy	(i) It increases computational cost (ii) It is not suitable for effective feature extraction
Dua et al. [25]	2021	CNN-GRU	(i) It has higher recognition accuracy (ii) It achieves superior classification performance	(i) The model cannot compromise the demand for accuracy and the computation cost. (ii) The network is hard to train
Khan et al. [26]	2021	Attention-induced multi-head CNN	(i) It effectively extracts salient discriminative features of the activities (ii) It achieved better accuracy	(i) Does not perform very well in highly complex activities (ii) Poor generalization capability of the network

**Table 2 sensors-23-04319-t002:** Proposed (SE-BiLSTM) model’s detailed structure.

Layer Name	Hyperparameter Settings
Convolutional	Activation = ReLU, Padding = Same, Strides = 1, Kernel Size = 5
Convolutional	Activation = ReLU, Padding = Same, Strides = 1, Kernel Size = 5
Dropout	Dropout Rate = 0.2
Max-Pooling	Padding = Same, Strides = 1, Pool Size = 8
Convolutional	Activation = ReLU, Padding = Same, Strides = 1, Kernel Size = 5
Convolutional	Activation = ReLU, Padding = Same, Strides = 1, Kernel Size = 5
Convolutional	Activation = ReLU, Padding = Same, Strides = 1, Kernel Size = 5
Dropout	Rate for dropout = 0.2
BiLSTM	Activation = Softmax, Recurrent Activation = Sigmoid, Dropout of Recurrent = 0.0
Flatten	-
Dropout	Rate for dropout = 0.2
Classification	Activation = Softmax

**Table 3 sensors-23-04319-t003:** Optimized hyperparameter settings using IFPOA.

Parameters	Range
Epochs	50
Batch size	64
Activation function	Sigmoid
Dropout rate	0.1
Embedding size	128
Learning rate	0.001
Loss	Categorical Cross Entropy
Output activation function	Softmax

**Table 4 sensors-23-04319-t004:** Proposed SE-BiLSTM-IFPOA using the MHEALTH dataset.

	Proposed Deep SE-BiLSTM-IFPOA Using the MHEALTH Dataset		
Activity	Precision (%)	Recall (%)	F1-Score (%)
Standing	100	94.97	97.52
Sitting	95.02	92.43	91.51
Lying down	93.14	95.65	94.45
Walking	92.06	96.98	92.57
Climbing stairs	87.12	88	91.87
Waist bends forward	96.76	91.94	94.07
Frontal elevation of arms	91.09	92.78	98.1
Knees bending	87.02	91.09	96.24
Cycling	98.98	84.12	87.98
Jogging	89.23	98.78	93.06
Running	91.32	87.98	98.03
Jumping front and back	93.65	97.07	91.93
Overall accuracy (%)			99.98

**Table 5 sensors-23-04319-t005:** Overall performance comparison with other deep learning-based models on the MHEALTH dataset.

Reference	Year	Model	Accuracy (%)	Computation Time (s)
Gholamrezaii et al. [28]	2021	2D CNN	99.88	0.4275
Jain et al. [29]	2022	CNN-LSTM	99.25	0.2541
Dua et al. [30]	2022	ICGNet	99.25	0.6257
Varshney et al. [31]	2022	Deep transfer learning (TL) model	98.63	0.5937
Nafea et al. [32]	2022	CNN-GRU	99.35	0.3203
Proposed Research		Deep SE-BiLSTM-IFPOA	99.98	0.2031

**Table 6 sensors-23-04319-t006:** Class-wise performance on the KU-HAR dataset of the proposed model for HAR.

Proposed Deep SE-BiLSTM-IFPOA Using the KU-HAR Dataset			
Activity	Precision (%)	Recall (%)	F1-Score (%)
Stand	99.00	98.82	98.92
Sit	96.1	94.2	95.25
Talk sit	94.52	97.23	95.45
Talk-stand	99.45	99.98	99.68
Stand-sit	98.62	99.23	98.95
Lay	97.82	98.62	98.2
Lay stand	98.85	98.23	99.8
Pick	99.99	99.78	99.7
Jump	100	99.99	99.99
Push-up	99.58	99.12	99.62
Sit-up	99.43	98.28	98.88
Walk	99.98	99.7	99.98
Walk-backward	99.98	99.42	99.63
Walk-circle	99.5	99.9	99.71
Run	99.83	99.9	99.8
Stair-up	99.82	99.73	99.86
Stair-down	99.91	99.62	99.72
Table tennis	100	99.97	99.93
**Overall accuracy (%)**			**99.76**

**Table 7 sensors-23-04319-t007:** Performance comparison results of the KU-HAR dataset.

Reference	Year	Model	Accuracy (%)	Computation Time
Dahou et al. [33]	2022	CNN-BAOA	96.8	0.3924
Kumar et al. [34]	2022	RecurrentHAR	98.98	0.6523
Luptáková et al. [22]	2022	Transformer method	99.2	0.5902
Proposed model		Deep SE-BiLSTM-IFPOA	99.76	0.2053

**Table 8 sensors-23-04319-t008:** Performance comparison results of the PAMAP2 dataset.

Reference	Year	Model	Accuracy (%)	Computation Time
Cheng et al. [24]	2022	Condconv CNN	94.01	0.4321
Jin et al. [35]	2021	CNN-D	94.72	0.543
Zhou et al. [36]	2022	MLWP model	95.93	0.5212
Li et al. [37]	2022	M-branch G-CNN	97.35	0.6142
Proposed Research		Deep SE-BiLSTM-IFPOA	99.54	0.3210

**Table 9 sensors-23-04319-t009:** Ablation Study.

Model	Average Accuracy (%)		
	MHEALTH	KU-HAR	PAMPA2
CNN-BiLSTM Subnet	95.97	93.78	95.09
Unidirectional HiHAR	97.09	95.11	96.29
Proposed (Deep SE-BiLSTM)	99.98	99.76	99.54

## Data Availability

Not applicable.

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
