# Peer review of "Deep SE-BiLSTM with IFPOA Fine-Tuning for Human Activity Recognition Using Mobile and Wearable Sensors"

_sensors, 2023, doi:10.3390/s23094319_

Round 1

Reviewer 1 Report

1.Line 53 "However, in real world applications, it's possible that HAR systems can't change to accommodate new users or situations, which results in low computational efficiency [18]."

Why HAR systems can't change to accommodate new users or situations

2.Line 66: BiLSTMs use both past and future information whereas the baseline LSTMs use only past information  once all-time series data have been collected.

How does BiLSTMs obtain future information?

Is "future information" the current information obtained in real-time?

3.Line 132: "To solve the difficulties of the existing HAR models, we propose a reliable and effective hybrid deep learning-based network model named Deep SE-BiLSTM-IFPOA model."

List which difficulties in Table 1 the authors solve.

4. Fig.1 The characters under Human Action Recognition are blurred.

5.Figure 3. Proposed model architecture.  Explain the differences compared with other similar architectures.

6.Were Eq 1 Eq 2and Eq3 referenced from Squeezenet or proposed by authors?

Were Eq 5-15 referenced from other work or proposed by authors?

Identify which equations are proposed by authors.

7. Add the cost of computation time when comparing with other work.

Compare the ratio of accuracy/cost of computation time for performance efficiencies. 

Reviewer 2 Report

In this work, the authors present a hybrid Deep SE-BiLSTM model to recognize the human activity. Experimental result is consisted with the proposed model, rending a competitive advantage compared to current methods, and the organization of this manuscript is good and adequate. The reviewer thinks that the manuscript can be accepted in addition to revise the word size (Figure 1,3 etc.) for well reading.

Reviewer 3 Report

The paper is well presented and easy to follow. Here are some suggestions to improve the paper:-

- In view of different hyper parameters are set throughout the proposed method, I would suggest authors to add an ablation study and justify how the hyper parameters are chosen. It is important especially when NN-based methods are used.

- The writings in all figures are not clear, please improve the clarity of them.

Round 2

Reviewer 3 Report

I am satisfied with this version, thank you for considering my previous comments.